# Segmentation of Low-Light Optical Coherence Tomography Angiography Images under the Constraints of Vascular Network Topology

**DOI:** 10.3390/s24030774

**Published:** 2024-01-25

**Authors:** Zhi Li, Gaopeng Huang, Binfeng Zou, Wenhao Chen, Tianyun Zhang, Zhaoyang Xu, Kunyan Cai, Tingyu Wang, Yaoqi Sun, Yaqi Wang, Kai Jin, Xingru Huang

**Affiliations:** 1School of Automation, Hangzhou Dianzi University, Hangzhou 310018, China; zhi.li@hdu.edu.cn (Z.L.); gaopeng.huang@hdu.edu.cn (G.H.); zbf@hdu.edu.cn (B.Z.); wenhao.chen@hdu.edu.cn (W.C.); tianyun.zhang@hdu.edu.cn (T.Z.); tingyu.wang@hdu.edu.cn (T.W.); syq@hdu.edu.cn (Y.S.); 2Department of Paediatrics, University of Cambridge, Cambridge CB2 1TN, UK; zx265@cam.ac.uk; 3Faculty of Applied Sciences, Macao Polytechnic University, Macao SAR 999078, China; p2317017@mpu.edu.mo; 4Lishui Institute, Hangzhou Dianzi University, Lishui 323000, China; 5College of Media Engineering, Communication University of Zhejiang, Hangzhou 310018, China; wangyaqi@cuz.edu.cn; 6Eye Center, The Second Affiliated Hospital, School of Medicine, Zhejiang University, Hangzhou 310027, China; jinkai@zju.edu.cn; 7School of Electronic Engineering and Computer Science, Queen Mary University of London, London E3 4BL, UK

**Keywords:** medical image processing, optical coherence tomography angiography, retinal vessel segmentation, retinal vein occlusion

## Abstract

Optical coherence tomography angiography (OCTA) offers critical insights into the retinal vascular system, yet its full potential is hindered by challenges in precise image segmentation. Current methodologies struggle with imaging artifacts and clarity issues, particularly under low-light conditions and when using various high-speed CMOS sensors. These challenges are particularly pronounced when diagnosing and classifying diseases such as branch vein occlusion (BVO). To address these issues, we have developed a novel network based on topological structure generation, which transitions from superficial to deep retinal layers to enhance OCTA segmentation accuracy. Our approach not only demonstrates improved performance through qualitative visual comparisons and quantitative metric analyses but also effectively mitigates artifacts caused by low-light OCTA, resulting in reduced noise and enhanced clarity of the images. Furthermore, our system introduces a structured methodology for classifying BVO diseases, bridging a critical gap in this field. The primary aim of these advancements is to elevate the quality of OCTA images and bolster the reliability of their segmentation. Initial evaluations suggest that our method holds promise for establishing robust, fine-grained standards in OCTA vascular segmentation and analysis.

## 1. Introduction

Optical coherence tomography angiography (OCTA) has emerged as a revolutionary non-invasive imaging technique, providing unparalleled visualization of retinal and choroidal microvasculature at capillary-level resolution. This exceptional capability of OCTA allows clinicians to assess the health of blood vessels, making it a critical tool for diagnosing and monitoring various ocular diseases, including diabetic retinopathy, age-related macular degeneration, and glaucoma [1,2,3]. By capturing depth-resolved perfusion information, OCTA can reveal subtle vascular changes associated with these conditions earlier and with finer precision compared to traditional angiographic methods. The segmentation of OCTA images, therefore, plays a vital role in the medical field, enabling detailed analysis and assessment of vascular health. However, the segmentation process often encounters challenges such as uneven luminance and inconsistent layering in the images, necessitating the development of sophisticated frameworks like BiSTIM for precise vascular segmentation in OCTA images.

Initial studies have also hinted at the utility of OCTA in detecting vascular biomarkers for neurological conditions like Alzheimer’s disease [4,5,6,7]. However, realizing the full potential of OCTA technology has been hampered by the challenges in analyzing the massive, multi-dimensional datasets it produces. OCTA scans consist of multiple cross-sectional B-scans at the same retinal location that are repeated over time. These B-scans are affected by the electronic noise of high-speed CMOS sensors, reducing the signal-to-noise ratio and resolution of OCTA images, and even causing OCTA image artifacts due to the non-uniform response [8]. Therefore, advanced algorithmic processing of the variations in the B-scans can extract blood flow information to reconstruct volumetric angiograms. Although native OCTA data is 3D, technical constraints often necessitate flattening OCTA images into 2D en face projections centered on vitally important retinal layers like the superficial capillary plexus (SCP) and deep capillary plexus (DCP) (Figure 1). This compression leads to a loss of depth information and obscures intricate three-dimensional relationships between interconnected vascular trees. Furthermore, precise manual segmentation of retinal vasculature from OCTA images is tremendously labor-intensive, time-consuming, and prone to human errors, underscoring the need for automated computational approaches.

In the study of retinal vascular diseases, many previous studies have shown that changes in the DCP are critical. These changes include ischemia and reperfusion in the DCP area, as well as the formation of new blood vessels, which have a significant impact on the final prognosis of the disease [9]. Due to the non-invasive and high-resolution characteristics of OCTA technology, we are able to visually observe the capillary network in the macular area. In addition, OCTA can perform independent analysis of the SCP and DCP. However, despite this, OCTA still faces some challenges in measuring Foveal Avascular Zone (FAZ) areas, such as data differences produced by equipment from different manufacturers and disputes over boundary segmentation methods. Currently, the use of OCTA to analyze the measurement data characteristics of the blood vessels in retinal vascular diseases has important clinical value. However, there is still a lack of sufficient research work on the automated diagnosis of retinal vein occlusion (RVO) disease and hemicentral retinal vein occlusion (HCRVO) disease based on OCTA images.

To address the limitations inherent in OCTA imaging and to harness its full clinical potential, our approach involves treating repeated scans as paired data. Despite the inherent differences in scanners, these repeated scans from the same eye typically exhibit similar anatomical features, albeit with varying artifacts and independent noise interference. To enhance the prior information in the segmentation structure, we have developed a Biological Information Signal Transduction Imaging Framework (BiSTIM), which employs a subpath structure constraint module. Additionally, we have designed a Proteomic-Inspired Topological Segmentation (PrIS-TS) module with a novel directional loss function. This module is specifically tailored to extract robust vascular representations in OCTA images, adeptly handling the complexity of different structural layers and branches. This segmentation module is particularly effective in encouraging the preservation of topological structures across various levels and branches. Furthermore, to mitigate common imaging artifacts such as projection shadows, we introduce a Bio-Luminescence Adaptation for Artifact Mitigation (BLAAM) module.

The rationale behind the selection of the BLAAM and STA modules is rooted in the specific challenges posed by OCTA imaging. The BLAAM module was conceived to address the issue of uneven brightness, a prevalent artifact in OCTA images, by eliminating noise associated with luminance irregularities. Similarly, the STA module was developed to ensure the accuracy of segmentation in the overarching capillary networks of OCTA images. This module is instrumental in maintaining consistency in the segmentation of primary and branching vascular structures, a critical factor in the accuracy of OCTA imaging.

The main contributions of this work include the following:We design the PrIS-TS module as part of BiSTIM. Deep topological structure supervision information and information interaction between different branches are utilized to enhance the topological structure information in the segmentation process to obtain the segmentation results.A subpath structure constraint (STA) module is developed to provide deep supervision signals to enhance the prior information in the segmentation structure.To mitigate imaging artifacts such as shadows and improve the clarity of OCTA images acquired under low-light conditions and various high-speed CMOS sensors, we introduce a bioluminescence-based technique.We collected 614 OCTA images from RVO and HCRVO. Experimental evaluation on two OCTA retinal vessel segmentation datasets, RVOS and OCTA-500, demonstrates the effectiveness of the proposed BiSTIM.

## 2. Related Works

### 2.1. Segmentation Methods in OCTA

Several previous studies have deeply explored the application of deep learning techniques in OCTA vascular segmentation. For instance, Morgan and his team [10] utilized the U-Net [11] architecture for the segmentation of vessels and retinal FAZ in surface SVP images from two scanners. Similarly, Mou and colleagues [12,13] proposed an attention module specifically designed for vascular segmentation and applied it to OCTA images. These innovative methods have opened up new possibilities for detailed and accurate vascular segmentation in OCTA images. Li and his team [14] proposed a unique method capable of directly outputting 2D vascular maps and FAZ segmentation from 3D OCTA images—a noteworthy innovation with potential implications for accurate OCTA-based diagnosis and treatment plans. The segmentation of 3D vessels from 3D OCTA volumes was investigated by Hu and his team [15]. Concurrently, a method for segmenting vessels from 2D OCTA images and estimating the depth information of segmented vessels for 3D vascular analysis was introduced by Yu and colleagues [16].

However, research on retinal vessel (RV) segmentation in OCTA images is relatively scarce due to the lack of publicly available OCTA image datasets with annotated vascular information. Despite this, the emergence of public datasets has sparked interest in deep learning-based RV segmentation methods [17,18]. For example, Ma and his team [18] developed a two-stage baseline network called OCTA-Net and applied it to their ROSE dataset, which was the first publicly available OCTA dataset with pixel-level annotations and manual RV segmentation and grading. Although some datasets are available, one of the major challenges faced in RV segmentation is the variation in thickness, especially at thin vessel endings, which display low contrast. In response to this challenge, Lee and Yeung [19] proposed a Supervised Vessel Segmentation Network (SVS-Net) for detecting varying sizes in retinal vein occlusion (RVO) OCTA images while preserving most of the vascular details and large non-perfusion areas. This method exemplifies the ongoing innovation in the field of OCTA image segmentation, especially in overcoming the challenges posed by the complexity of these images.

### 2.2. Direction Segmentation Low-Light Scenes

In topology studies, pixel connectivity is employed to depict the association between adjacent pixels. This is a classical image processing technique, extensively used for characterizing topological properties. In deep learning-based image segmentation, connectivity has found novel applications. Segmentation networks based on connectivity utilize connection masks as labels. These masks are defined as eight-channel masks, where each channel represents the association of a pixel in the original image with its neighboring pixel in a specific direction. These neighboring pixels belong to the same category. Connection masks were first introduced and applied in image segmentation. This concept was subsequently expanded by other researchers and integrated into their work, including showcasing the bidirectional nature of pixel connectivity in saliency detection and cross-modal connection data fusion in simulated radar videos. Meanwhile, the effective modeling of connectivity has been demonstrated in various applications, such as remote sensing segmentation, path planning, and medical image segmentation. Despite significant advancements in this domain, we found that the rich directional information in connection masks is yet to be fully leveraged.

In another study, the researchers constructed a new dataset, called the LOL dataset [20], by adjusting exposure times. They also designed RetinexNet, which occasionally produces unnatural enhancement effects. The authors of KindD [21] addressed this issue by introducing certain training losses and adjusting the network structure. In [22], the authors proposed Deep-UPE, where an illumination estimation network was defined to enhance low-light inputs. Another study [23] proposed a recursive network and a semi-supervised training strategy. In [24], the authors proposed EnGAN, designing a generator that focuses on enhancement under unpaired supervision. The authors of SSIENet [25] built a decomposition framework for the simultaneous estimation of illumination and reflectance. In [26], the authors proposed ZeroDCE, heuristically constructing a curve with learned quadratic parameters. Recently, Liu et al. [27] established a Retinex-inspired unfolded framework through an architecture search. Despite these deep networks being meticulously crafted, they are often unstable, struggling to consistently deliver superior performance, especially when facing unknown real scenes and blurred details.

## 3. Method

Our proposed Biological Information Signal Transduction Imaging Framework (BiSTIM) comprises two main modules: the Proteomic-Inspired Topological Segmentation (PrIS-TS) module and the Bio-Luminescence Adaptation for Artifact Mitigation (BLAAM) module. As illustrated in Algorithm  1, this demonstrates the specific algorithmic process for OCTA image segmentation. As illustrated in Figure 2, this system takes two specific layers from OCTA images—the superficial capillary plexus (SCP) and deep capillary plexus (DCP)—as inputs. These two layers of microvascular networks are distributed in different areas of the retina and are responsible for providing nutrients. It is noteworthy that due to the structural and functional differences between the SCP and DCP, they may show varying degrees of damage in certain eye diseases such as retinal vein occlusion (RVO) and the highly complex hemicentral retinal vein occlusion (HCRVO). This poses specific challenges and problems for OCTA image analysis, including but not limited to the accurate segmentation of vascular networks, the detection of blood flow changes, and optical interference and image distortion. In addressing these challenges, our framework specifically aims to segment two classes of pixels: those representing the vascular structures and those representing the background. This distinction is crucial for the accurate analysis and diagnosis of the aforementioned conditions.

To address these problems, our segmentation module uses a method based on a deep supervision feature module to extract multi-scale deep features from both the SCP and DCP layers and generate topological structure supervision information for the next stage. Furthermore, our directional proteomics pathway sequencer module performs deep structural feature extraction and topological structure supervision signal generation for these two retinal layers, thereby enhancing the comprehensive analysis of retinal-layer lesions and providing more accurate and detailed segmentation and diagnosis of vascular networks. Figure 2 shows the modules comprising our BiSTIM and the flow of the mask generation.
**Algorithm 1** Algorithm for OCTA image segmentation1:**Input:** OCTA image *I*2:**Output:** Segmentation of retinal vessels and capillaries3:Get feature maps via ResNet50 as $F_{l}$4:Upsample the last feature map $F_{5}$ as $F_{\mathrm{aux}}$5:Slice the channel direction and excitate the subpath topology through the STA module to generate a new feature map $F_{\mathrm{STA}}$ as a deep signal6:Supervise the genomic signal feature interpreter to segment retinal vessels and capillaries7:Equalize the luminosity of the segmentation results and remove artifacts and noise through the BLAAM module to segment retinal vessels and capillaries

### 3.1. Proteomic-Inspired Topological Segmentation (PrIS-TS) Module

OCTA images can reveal complex structures of the retinal FAZ and capillary networks on a microscopic scale. These capillary networks mainly consist of vessels and fine-branching vessels, with more intricate branching structures extending from these primary vessels. Many existing studies overlook a salient feature of OCTA images: their relatively stable topological structure. To fill this gap in the research, we were inspired by proteomics and focused our investigation on the stable topological structures at different levels within OCTA images.

We have developed a Proteomic-Inspired Topological Segmentation (PrIS-TS) module. This module accepts OCTA images at various levels as input and incorporates a dedicated multi-branch structure to continually reinforce the topological properties of the capillary networks. More specifically, this module comprises two main branches: the directional proteomic pathway sequencer branch and the genomic signal feature interpreter branch. The pathway sequencer branch is responsible for extracting deep features from multiple scales. Simultaneously, the pathway sequencer branch interacts with the feature interpreter branch to continually reinforce the topological stability of capillary networks during the segmentation process.

#### 3.1.1. Directional Proteomic Pathway Sequencer

In complex image segmentation tasks, the extraction of structural priors and directional information is a crucial step. The method we propose addresses the shortcomings of traditional image segmentation methods in this regard by extracting structural priors from the intermediate features of the encoder and compressing and obtaining directional information through directional embedding. This novel approach offers a new solution for image segmentation.


**Extraction of Structural Priors.**


In our method, the extraction step of structural priors involves using the channel directional information in the connectivity mask to directly acquire techniques for unique directional embedding. This process can roughly supervise the intermediate features of the encoder and compress the channels of the auxiliary connectivity output. As shown in Figure 3, we represent the encoder’s output as $F_{l}$, where *l* denotes the $l^{th}$ layer of the encoder. Specifically, $F_{l}$ is the deep feature extracted by the encoder, which comprises five layers. We utilize a pretrained ResNet50 as the encoder, and the output of this encoder is denoted as $F_{l}$, with *l* representing the layer number in the encoder, where the maximum value of *l* is 5. In the implementation of our model, we upsample the final encoder output $F_{5}$ to the input size to obtain the preliminary output, referred to as $F^{a}ux$. During the calculation of the loss, this preliminary output is supervised to learn the connectivity masks. The structural priors are extracted by the STA module, which consists of direction prior extraction, channel-wise slicing, and subpath topology excitation.

From the channels of $F^{a}ux$, we can obtain rich and unique directional information. We then apply Global Average Pooling (GAP) to $F^{a}ux$ to compress the size. Next, we map the vector to the same dimension as the latent feature map $F_{L}\in R^{C_{FL}\times H_{FL}\times W_{FL}}$ using a 1 × 1 convolution kernel $K_{1}$:(1)$$GAP\left(F^{a}\right)=\frac{1}{H\times W}\sum_{i=1}^{H}\sum_{j=1}^{W}F^{a}\left(i,j\right),$$
(2)$$I^{a}ux=\sigma (K_{2}\delta \left(K_{1}GAP\left(F^{a}ux\right)\right).$$
where *H* and *W* denote the height and width of the feature map, respectively. $K_{1}\in R^{C_{F_{5}}\times C_{a}}$, $C_{a}$ is the channel number of $F^{a}ux$, and δ is the ReLU activation. Then, as shown in Equation (Equation 2), we re-encode using a 1 × 1 convolutional kernel $K_{2}$ and apply the sigmoid gating function σ to normalize the projection vector.

Since $I^{a}ux$ contains rich element direction information, we define it as a directional prior. This approach assists our model in better understanding the directional characteristics in the image, thereby improving the model’s performance in complex image segmentation tasks.

**Channel Direction Slicing.** In dealing with complex image segmentation problems, we propose a novel approach—channel direction slicing and subpath topology excitation. In order to decouple the classification and direction subspaces in the hidden layers as early as possible, we employ CDS to slice the latent features ($F_{5}$) and direction priors ($I^{a}ux$) into eight parts. Specifically, we represent the $t^{th}$ slice as $F_{5}^{t}$ and $F^{a}ux^{t}$.

Then, for each pair of these feature-embeddings slices, we construct a subpath. Within each subpath, we pass the feature slice $F_{5}^{t}$ through a spatial and channel attention module to capture the long-range and channel-wise dependencies, resulting in $F_{5}^{t^{\prime }}$. Then, we perform an element-wise multiplication of $I^{a}ux^{t}$ with $F_{5}^{t^{\prime }}$ channel-wise to selectively highlight or suppress features with specific directional information. We then re-encode the output using a 1 × 1 convolutional kernel $K_{3}^{t}$ and consider this as the residual output:(3)$$F_{STA}^{t}=K_{3}^{t}\left(I^{a}ux^{t}\cdot F_{5}^{t^{\prime }}\right)+F_{5}^{t}.$$

Finally, we stack all subpath outputs ($F_{STA}^{t}$) together and re-encode them, resulting in a new feature map $F_{STA}$.

Due to the slicing operation, each slice group will contain only partially complete features. However, differences will arise due to the different reductions in the distinctive contextual information in the direction and classification features. Specifically, $I^{a}ux$ is a highly distinguishable directional embedding as it is a low-level linear combination of unique directional features. Thus, channel direction slicing will result in a significant reduction in the directional information in each slice. However, $F_{5}$ contains a set of high-level but less distinguishable classification features, among which variations are usually small. Hence, high channel classification correlation and redundancy exist in $F_{5}$. Therefore, channel direction slicing will result in a smaller reduction in the highly distinguishable classification features in $F_{5}^{i}$.

Our approach unevenly partitions the directional and classifying features within each subpath, thereby emphasizing the dominant features. This facilitates the learning of class-specific information on channels where a smaller amount of unique directional information is stressed while also learning the different unique directional information between subpaths. Once the subpaths are stacked, the directional information naturally decouples from the original latent space and embeds into the channels. This innovative design method not only enhances the accuracy of image segmentation but also provides a solution based on stable topological structures. It improves model performance and expands the potential applications in fundus imaging and other medical imaging scenarios.

#### 3.1.2. Genomic Signal Feature Interpreter

In the process of feature decoding, many subtle attributes may be lost. To address this issue, we have designed a module named the genomic signal feature interpreter. As shown in Figure 4, the feature interpreter primarily consists of two parts—the dual-stream branch and the topology decoder—each serving a specific function. The dual-stream branch can gradually optimize the segmentation of trunks and capillaries in OCTA images by combining the feature map from the pathway sequencer’s feature map and the deep topology information from the STA module. Finally, the multiple outputs of the dual-stream branch are decoded by the topology decoder to obtain segmentation results with clear trunks and optimized details.

To optimize the segmentation of the main blood vessels and capillaries in the SCP and DCP layers using deep topology supervision signals, we designed a feature stream comprising a spatial stream and a metablock combined with a perception block, which are responsible for embedding compression and manifold projection, respectively. First, the metablock decodes the depth topological structure information $F_{STA}^{t}$ from the STA module to obtain $d_{t}$. Then, the perception block embeds $d_{t}$ into $f_{l}$ retrieved from the pathway sequencer through manifold projection within the perceptual structure, thereby realizing multi-level supervised segmentation of microvascular structures and optimizing the multi-faceted segmentation of the trunk and detail branches. The computational process is delineated as follows:(4)$$d_{t}^{\prime }=\sigma \left(F_{l}\cdot GAP\left(F_{STA}^{t}\right)\right).$$

To obtain a final segmentation result that combines multiple levels of supervision signals, we employ a simple topology decoder. Given the presence of multi-level blood vessels and complex structures in OCTA, the feature interpreter is designed as a multi-level structure dual-stream network, which achieves multi-layer segmentation of the SCP and DCP in RVO and HCRVO diseases. Through the use of deep topology result signals, the optimization of segmentation results is realized under a multi-branch structure.

### 3.2. Bio-Luminescence Adaptation for Artifact Mitigation (BLAAM)

During the acquisition process of OCTA images by high-speed CMOS sensors, issues such as artifacts and image blurring often occur, leading to significant discrepancies in the images in terms of contrast, brightness, and other aspects. In particular, over-exposed or under-lit images generally degrade image quality, thus adding extra complexity to image annotation and automatic segmentation tasks.

We propose a new method called Bio-Luminescence Adaptation for Artifact Mitigation (BLAAM) to address the difficulties in image segmentation caused by uneven brightness. This method has not only demonstrated significant improvements in capillary segmentation but has also yielded noticeable enhancements in visual effects. This technique fills the gap in existing research on the issue of uneven brightness in OCTA images.

BLAAM incorporates a crucial module, namely the luminescent distribution integration module. This module can collect the luminous distribution in the image and adaptively adjust the image to a standardized acquisition effect based on these statistics. Furthermore, we designed an auto-regulatory calibrated interface, which can calculate the brightness difference in OCTA images in real time across different time periods.

Consequently, BLAAM offers clinicians a practical tool for overcoming the diagnostic challenges arising from image quality issues, thereby improving the accuracy and consistency of diagnoses.

#### 3.2.1. Luminescent Distribution Integration Module

The issue of image quality under low-light conditions and various sensors is a common problem in OCTA datasets. According to the classic Retinex theory, we know there is a relationship between the observed image *y* under low-light conditions and the expected clear image *z*, which can be expressed as y=z⊗x, where *x* represents the illumination component. In this relationship, illumination is considered the core constituent that needs to be primarily optimized in low-light image enhancement. Following the Retinex theory, we can obtain the enhanced output image by eliminating the estimated illumination component. In this process, our method is inspired by [27,28,29], where phased optimization of the illumination component was performed. We base our approach on a mapping $P_{\theta }$ with a parameter θ and approach this task progressively, where the basic unit can be represented as:(5)$$R\left(x^{t}\right):\left\{\begin{array}{c} e^{n}=P_{\theta }\left(x^{n}\right),x^{0}=y,\\ x^{n+1}=x^{n}+e^{n}. \end{array}\right.$$

Here, $e^{n}$ and $x^{n}$ represent the residual term and the illumination at the *n*-th stage (n=0,...,N−1), respectively. The mapping $P_{\theta }$ is a parameterized operator that learns a simple residual representation $e^{n}$ between the illumination and low-light observation.

This process is inspired by the understanding that the illumination and low-light observation are similar or have linear connections in most areas. By learning a residual representation instead of adopting a direct mapping, $P_{\theta }$ substantially reduces the computational difficulty, enhancing performance and stability, particularly in exposure control. We adopt a weight-sharing mechanism in $P_{\theta }$, using the same architecture P and weights θ at each stage.

We could directly learn an enhancement model given the training data and loss function. However, cascading multiple weight-sharing modules slows down inference. The goal of each module is to output an image close to the target. Ideally, the first module alone could satisfy this. Later modules produce redundant outputs. Therefore, during testing, we can speed up inference by using just the first module. Our method enhances low-light OCTA images while also improving inference speed and stability through weight sharing and stepwise optimization. This will benefit subsequent analysis and clinical use of retinal images.

#### 3.2.2. Auto-Regulatory Calibrated Interface

To define a module that allows computational results at each stage to converge to the same state, we must first recognize that the input for each phase originates from the previous one, with the first phase’s input naturally being our low-light observation.

An intuitive idea would be to establish a connection between the input of each stage (except for the first one) and the low-light observation (i.e., the input of the first stage), thus indirectly exploring the convergence between each stage. To achieve this goal, we introduce a self-calibration module *h* and add it to the low-light observation to represent the difference between the input of each stage and the input of the first stage. More specifically, the self-calibration module can be expressed as:(6)$$T\left(x^{n}\right):\left\{\begin{array}{c} z^{n}=y⊘x^{n},\\ h^{n}=H_{\varphi }\left(z^{n}\right),\\ c^{n}=y+h^{n}, \end{array}\right.$$
where n≥1, $c^{n}$ is the transformed input at each stage and Hφ is the parameterized operator we introduced with learnable parameters φ. The operator Hφ plays a crucial role in our framework. It is designed to adaptively adjust the input of each stage based on the learned parameters φ. This adjustment is achieved through a process of parameterized transformation, where Hφ applies a set of learned transformations to the input $z^{n}$, thereby generating the self-calibration term $h^{n}$. This term is then used to modify the input of the subsequent stage, ensuring that the input is optimally adjusted for each stage of the process. Furthermore, the basic unit transformation at the *n* stage (n≥1) can be written as $R\left(x^{n}\right)\to R\left(T\left(x^{n}\right)\right)$. The introduction of $H_{\varphi }$ allows for a more dynamic and adaptive approach to handling the variations in illumination and other factors in low-light OCTA images, significantly enhancing the effectiveness of our method.

In practice, our constructed self-calibration module integrates physical principles to gradually correct the input at each stage, thereby indirectly influencing the output at each stage. This framework combines physical principles with deep learning algorithms, aiming to improve the efficiency and stability of algorithms while ensuring image quality. Our experimental results have also confirmed the effectiveness of this design, reducing the impact of noise, such as artifacts and uneven brightness, on segmentation during the OCTA acquisition process.

## 4. Experiments and Results

### 4.1. Dataset and Metrics

We tested our model on two datasets: the RVOS dataset, collected by the Second Affiliated Hospital of Zhejiang University, and the widely recognized OCTA-500 [30] dataset. The RVOS dataset is particularly notable for its inclusion of OCTA images captured in low-light conditions, a prevalent challenge in medical image segmentation. The OCTA-500 dataset, on the other hand, offers a diverse range of scenarios and multiple acquisition layers, making it ideal for assessing the generalizability of our model. These datasets collectively provide a comprehensive testbed for our model, demonstrating its applicability in various clinical scenarios and its ability to handle common challenges in the field.

**RVOS:** This dataset includes 454 training images and 160 test images, with a mix of 140 HCRVO and 167 RVO images. The images were captured using high-speed CMOS to record the light-intensity signals reflected and scattered back by tissues at different depths. The inclusion of low-light condition images in RVOS makes it a valuable dataset for testing the robustness of our model in challenging imaging scenarios.**OCTA-500:** All 200 subjects (No. 10301-No. 10500) with 3mm×3mm SVP scans from the OCTA-500 dataset [30] were included in our experiments. The data were collected using a commercial 70 kHz SD-OCT (RTVue-XR, Optovue, Fremont, CA, USA). We used the maximum projection map between the internal limiting membrane (ILM) and the outer plexiform layer (OPL) because it was used for vessel delineations. We followed the same training, validation, and testing split as in [30] (No. 10301-10440 for training; No. 10441-10450 for validation; and No. 10451-10500 for testing).

The details of the datasets are shown in Table 1. Our selection of these datasets, namely the RVOS dataset collected by the Second Affiliated Hospital of Zhejiang University (referred to as ROVS in some contexts) and the publicly available OCTA-500 dataset, was driven by their ability to present a wide range of imaging conditions and challenges. This selection enabled us to comprehensively evaluate the generalization capabilities of our model in a clinical context. The proposed model was rigorously tested on both datasets, and the segmentation results achieved state-of-the-art (SOTA) performance. This demonstrates that our model possesses robust generalization capabilities across different datasets, making it suitable for most current OCTA image segmentation tasks.

To evaluate the performance of the vessel segmentation algorithms, the following metrics were calculated between the manual delineation and the segmentation results produced by each algorithm:Recall, specificity, IoU, and Dice: $DSC=\frac{2\times TP}{2\times TP+FP+FN}$;Area under the ROC curve: AUC;Accuracy: $ACC=\frac{TP+TN}{TP+TN+FP+FN}$;Kappa score: $KAPPA=\frac{ACC-pe}{1-pe}$;False discovery rate: $FDR=\frac{FP}{FP+TP}$;G-mean score: $GMEAN=\sqrt{sensitivity\times specificity}$;Dice coefficient: $DSC=\frac{2\times TP}{FP+FN+2\times TP}$.

TP, TN, FP, FN represent the true positives, true negatives, false positives, and false negatives, respectively, and $pe=\frac{\left(TP+FN)(TP+FP)+(TN+FP)(TN+FN\right)}{{\left(TP+TN+FP+FN\right)}^{2}}$. Sensitivity and specificity are computed as $\frac{TP}{TP+FN}$ and $\frac{TN}{TN+FP}$, respectively. These metrics are also reported in [18]. All the *p*-values reported were computed using a paired, two-sided Wilcoxon signed-rank test (null hypothesis: the difference between paired values comes from a distribution with a zero median).

### 4.2. Implementation Details

Our model was implemented using PyTorch, and all the networks were trained using the Adam optimizer. The learning rate was set at $1\times 10^{-3}$, and the weight decay was established at $1\times 10^{-6}$. The Adam optimizer, which combines the advantages of RMSProp and Momentum, was well suited for our network’s requirements. We utilized the Adam optimizer with the parameters $\beta_{1}=0.9$, $\beta_{2}=0.999$, and $\epsilon =10^{-8}$. The minibatch size during training was set to 8, and the learning rate was initialized at $10^{-4}$. We set the number of training epochs to 1000 to ensure comprehensive learning and optimization of the network. The parameters θ in Equation (Equation 5) and φ in Equation (Equation 6) represent the learnable parameters within the respective modules of our framework. The parameter θ in the mapping Pθ is critical for the phased optimization of the illumination component, as it governs the degree of residual learning at each stage. Similarly, the parameter φ in the self-calibration module Hφ is essential for adjusting the input at each stage based on the learned self-calibration. The values of θ and φ are learned during the training process, allowing the model to adaptively enhance its performance on the OCTA image datasets. Training ceases when the validation loss no longer decreases, employing an early-stopping strategy to effectively prevent overfitting on the training set, provided a suitable validation dataset is available. Otherwise, the number of training epochs is determined empirically. For instance, the OCTA-500 dataset possesses its own validation dataset, whereas for our other experiments, the number of training epochs is determined based on empirical evidence. All experiments were conducted on a single 4090 graphics card, ensuring computational efficiency and result reproducibility. Table 2 shows the time required to generate the outputs for each module of our network. Currently, the source code for this work remains proprietary while under review for potential commercialization. Upon completion of the review, we will consider making the source code publicly available to facilitate the replication of our experiments by other researchers and further advancement in the field.

### 4.3. Performance Comparison and Analysis

Our proposed framework demonstrated superior performance across multiple OCTA image datasets, as evaluated through quantitative metrics and visual inspection. As shown in Table 3, BiSTIM achieved optimal results across all metrics and regions. We compared our proposed method to several state-of-the-art architectures including U-Net [11], R2U-Net [31], attention-gated R2U-Net [32], U-Net+CBAM [33], U-Net+SK [34], SegNet [35], ENet [36], PSPNet [37], DeepLabV3+ [38], GCN [39], UNet++ [40], UNet 3+ [41], Frago [42], and IPN+ [43].

**Qualitative comparison:** We first evaluated the performance of the proposed method on the RVOS dataset against a variety of SOTA methods using the aforementioned evaluation metrics. As shown in Table 4, except for recall and specificity, our proposed BiSTIM method exhibited the best performance in both the SCP and DCP classes. Although UNet3+ and U-Net achieved the best results in the recall and specificity metrics, they failed to accurately measure the model’s segmentation, and there was a serious imbalance between U-Net and UNet3+ in the assessment of microvessels and background classes. For example, UNet3+t achieved the optimal results in recall (0.89195), but its specificity (0.90247) was much lower compared to U-Net (0.99016). The Kappa and GMEAN are two metrics that can reflect comprehensive segmentation quality and better reflect the model’s overall performance on the data. However, the proposed model achieved optimal results in both the mean value of Kappa (0.81138) and the GMEAN (0.91074). This indicates that BiSTIM effectively combined vascular structure segmentation in the foreground with multi-level deep topological structure information and was able to distinguish microvessel classes from background classes.

In addition, the models demonstrating effective segmentation adopted skip-link operations. For example, models such as UNet++ and Frago achieved partial segmentation of microvessels in OCTA images through the supervision of more complex multi-scale depth information. Therefore, the feature interpreter we designed achieved optimal results in the Dice, Kappa, and GMEAN metrics through multi-level supervision of deep topology. Compared to Frago, the proposed model exhibited a 0.77% improvement in the Dice metric, which proves that the design of BiSTIM is consistent with the OCTA segmentation task compared to other methods.

**Quantitative comparison:** The same conclusion is supported by the OCTA segmentation results in Figure 5. The performance of many of the classic segmentation models in the OCTA microvessel segmentation task was poor in several aspects. For example, in the segmentation of the main trunk of microvessels and the detailed segmentation of branches, most methods could not identify the main branches of blood vessels, and breakpoint problems occurred in the segmentation of veins and capillaries. The proposed method was able to segment the details in both the trunk structure and the veins of branch vessels.

To further validate the proposed method, we evaluated its performance on the publicly available OCTA-500 dataset. As shown in Table 4, the BisTIM achieved a mean Dice coefficient of 0.8865, IoU of 0.79707, and accuracy of 0.91317 on this dataset. This surpassed the published results of state-of-the-art methods, including U-Net + AG (IoU 0.75286), UNet++ (IoU 0.76794), and Frago (IoU 0.75857). And it achieved great improvements in accuracy (2.7%), IoU (3.85%), and Dice (2.61%) compared to the Frago model. These results provide additional evidence that the proposed model generalized well across the OCTA datasets. The extensive quantitative benchmarking on OCTA-500 against the state-of-the-art methods further validates the strengths of the proposed approach for enhanced OCTA image analysis. It proves that the model’s multi-level decoding and enhancement of deep topological structure information can improve generalization and robustness.

### 4.4. Ablation Studies

#### 4.4.1. Effectiveness of STA Module

In order to verify the impact of the depth topology signal extracted by the STA module on the final segmentation result, we conducted ablation experiments on the RVOS dataset. As shown in Table 5, the proposed method surpassed other methods, like concatenation (Cat) and attention feature fusion (AFF), and achieved superior results in almost all metrics. Compared with AFF, the accuracy, IoU, and Dice metrics increased by 1.2%, 2.1%, and 1.5%. The improvement in segmentation proves the impact of the proposed deep topology method on the microvessel segmentation task at different levels in OCTA. The results highlight the benefits of topological relationships for combing OCTA layers in a synergistic manner.

#### 4.4.2. Effectiveness of BLAAM Module

**Qualitative comparison:** During the OCTA image acquisition process, problems such as artifacts and image blur caused by various high-speed CMOS are often encountered, further affecting segmentation quality. Therefore, in order to verify that our model can improve segmentation on data with uneven brightness, we compared its impact under different loss functions. As shown in Table 6, the BLAAM module achieved the best results in accuracy, IoU, and Dice compared to the other models, showcasing improvements of 3.6%, 4.9%, and 3.2%, respectively, compared to the sub-optimal method. These results show that the proposed BLAAM model can standardize data with uneven brightness, thereby optimizing the segmentation capability of the model and reducing the impact of noise caused by the acquisition process.

**Quantitative comparison:** The same conclusion is supported by the OCTA segmentation results in Figure 6. It can be observed that under the supervision of the BLAAM module, various trunks and capillaries were segmented in both the SCP and DCP layers, whereas the other methods could not distinguish between trunks and capillaries, resulting in a loss of details of the capillaries. However, $L_{CEDICE}$ obtained objective segmentation results in both qualitative and quantitative aspects. $L_{CEDICE}$ used cross-entropy to constrain the global structure and further controlled the details using the Dice loss function. This demonstrates the importance of constraining segmentation results through multi-level structures and further proves the crucial role of our BLAAM and STA modules within the overall BiSTIM framework.

## 5. Conclusions

In summary, this study enhances OCTA image segmentation, addressing artifact noise and complex vascular structures. BiSTIM comprises the PrIS-TS, STA, and BLAAM modules, showing superior performance in handling complex structures and artifacts on the RVOS and OCTA-500 datasets. Its utility in clinical applications, especially in detecting RVO and HCRVO diseases, is also demonstrated.

Although BiSTIM advances OCTA segmentation, future research will focus on refining the algorithms for patient-specific variability in OCTA images and integrating machine learning with clinical expertise for more personalized diagnostics. Exploring real-time OCTA image processing and extending BiSTIM to other imaging modalities are promising directions. Additionally, ethical considerations and patient privacy in AI-based systems in healthcare are crucial.

## Figures and Tables

**Figure 1 sensors-24-00774-f001:**
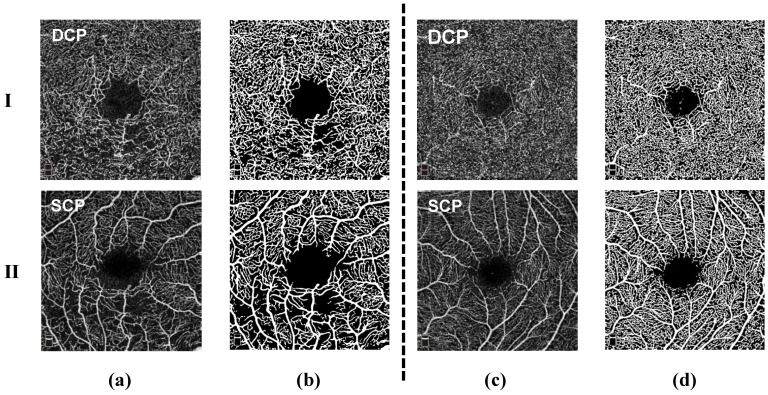
Comparison of OCTA images: (**a**) Rows 1 and 2 represent the DCP and SCP layers from RVO disease and their corresponding masks (**b**). (**c**) Rows 1 and 2 represent the DCP and SCP layers from HCRVO disease and their corresponding masks (**d**).

**Figure 2 sensors-24-00774-f002:**
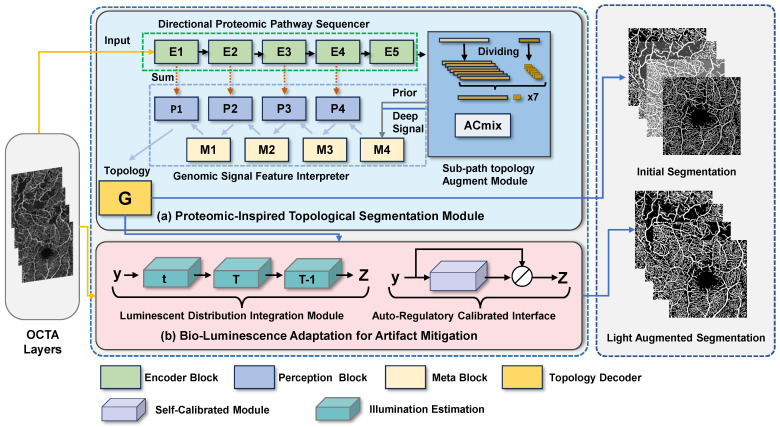
Illustration of our proposed BiSTIM. The PrIS-TS in (**a**) uses deep topological structure supervision information and information interaction between different branches to enhance the topological structure information in the segmentation process to obtain the segmentation results. The BLAAM in (**b**) can standardize the input OCTA image and optimize segmentation through adaptive photometric residual learning.

**Figure 3 sensors-24-00774-f003:**
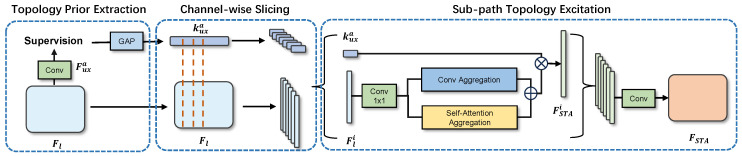
Illustration of the proposed STA module, which includes three steps: direction prior extraction, channel-wise slicing, and sub-width slicing.

**Figure 4 sensors-24-00774-f004:**
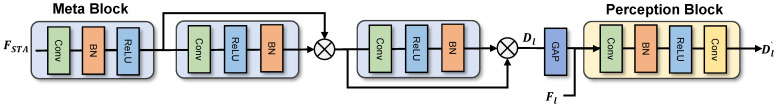
Illustration of the proposed dual-stream module.

**Figure 5 sensors-24-00774-f005:**
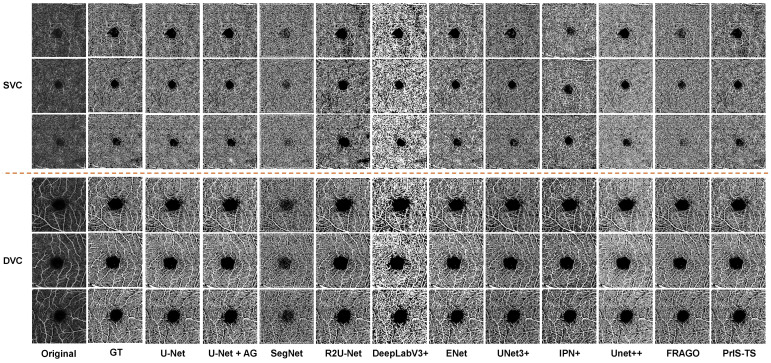
Visual comparison of microvessel segmentation between the proposed method and state-of-the-art methods on RVOS. The upper three rows are the data in the SVC class, and the lower three rows are the data in the DVC class. The segmentation of the model can be evaluated by comprehensively segmenting the trunk and branch structures of the reticular microvessels.

**Figure 6 sensors-24-00774-f006:**
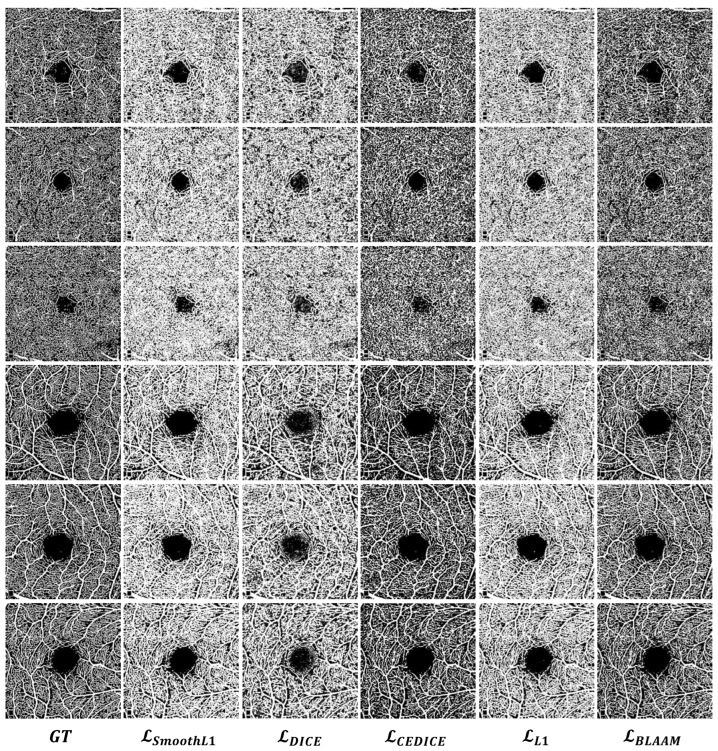
Visual comparison of monocular ophthalmic image segmentation between the proposed method and state-of-the-art methods.

**Table 1 sensors-24-00774-t001:** Details of the datasets used.

	OCTA-3M	RVOS
Number	200	614
Resolution	304×304	614×614
Image type	SVC	SVC, DVC
Annotation	pixel level	pixel level
Disease type	-	RVO, HCRVO

**Table 2 sensors-24-00774-t002:** The time required to run the different modules within the proposed framework (in us). It can be seen that each module runs very efficiently.

Module	Pathway Sequencer	STA Module	Feature Interpreter	Decoder	BLAAM Module
Time (µs)	0.3576	0.9537	0.4174	0.4768	0.7153

**Table 3 sensors-24-00774-t003:** Quantitative results of the proposed BiSTIM and previous SOTA models on the RVOS dataset. The IoU and Dice metrics reflect the quality of the average segmentation results, whereas the Kappa and GMEAN objectively evaluate the comprehensive segmentation quality of the mapping model under different distributions of data, which are important in OCTA image segmentation. Figures in bold indicate the best results.

Model	Recall	Specificity	Accuracy	IoU	Dice	Kappa	GMEAN
U-Net [11]	0.75383	**0.99016**	0.87466	0.74629	0.85293	0.74474	0.86294
U-Net + AG [44]	0.77178	0.98480	0.88394	0.75916	0.86178	0.76168	0.87097
SegNet [35]	0.81311	0.80561	0.80959	0.56305	0.72010	0.57808	0.80728
R2U-Net [31]	0.70778	0.90013	0.80622	0.62920	0.77141	0.60290	0.79595
DeepLabv3+ [38]	0.66382	0.84648	0.76596	0.56371	0.71964	0.51783	0.74807
PSPNet [37]	0.48245	0.83922	0.58582	0.42958	0.59870	0.20680	0.63495
ENet [36]	0.79958	0.88435	0.85193	0.67881	0.80759	0.68453	0.84009
GCN [39]	0.66387	0.79017	0.74454	0.50129	0.66640	0.45591	0.72311
UNet3+ [41]	**0.89195**	0.90247	0.90022	0.76309	0.86497	0.78515	0.89590
IPN+ [43]	0.89606	0.89936	0.89969	0.76105	0.86364	0.78370	0.89644
UNet++ [40]	0.77575	0.98719	0.8874	0.76483	0.86476	0.76878	0.87392
Frago [42]	0.87397	0.92738	0.90814	0.78399	0.87819	0.80362	0.89866
**BiSTIM**	0.86386	0.94036	**0.91118**	**0.79581**	**0.88580**	**0.81138**	**0.91074**

**Table 4 sensors-24-00774-t004:** Evaluation of SOTA methods in terms of OCTA-500 segmentation. Figures in bold indicate the best results.

Model	Recall	Specificity	Accuracy	IoU	Dice
U-Net	0.74875	**0.99026**	0.87390	0.74125	0.84943
U-Net + AG	0.76458	0.98594	0.88210	0.75286	0.85742
Unet++	**0.88648**	0.91008	0.90374	0.76794	0.86794
Frago	0.76900	0.98774	0.88600	0.75857	0.86046
**BiSTIM**	0.86097	0.94443	**0.91317**	**0.79707**	**0.88650**

**Table 5 sensors-24-00774-t005:** Evaluation of different fusion methods in terms of RVOS segmentation. Figures in bold indicate the best results.

Fusion Method	Recall	Specificity	Accuracy	IoU	Dice
Cat	0.77179	0.88067	0.83769	0.66154	0.79517
AFF [45]	0.82319	**0.95276**	0.89857	0.77399	0.87077
**STA**	**0.86386**	0.94036	**0.91118**	**0.79581**	0.88580

**Table 6 sensors-24-00774-t006:** Evaluation of different loss functions in terms of RVOS segmentation. Figures in bold indicate the best results.

Loss	Recall	Specificity	Accuracy	IoU	Dice
$L_{smoothL1}$	0.44308	**0.99987**	0.50747	0.44307	0.61177
$L_{DICE}$	0.72692	0.92355	0.82637	0.66887	0.80044
$L_{CEDICE}$	0.84813	0.92515	0.89541	0.76264	0.86457
$L_{L1}$	0.75383	0.99016	0.87466	0.74629	0.85293
$L_{\mathrm{BLAAM}}$	**0.86386**	0.94036	**0.91118**	**0.79581**	**0.88580**

## Data Availability

ROSE dataset link: https://imed.nimte.ac.cn/dataofrose.html (accessed on 2 January 2024); OCTA-500 dataset link: https://ieee-dataport.org/open-access/octa-500 (accessed on 2 January 2024); project code link: https://github.com/RicoLeehdu/BiSTIM (accessed on 2 January 2024).

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
