# Peer review of "Segmentation of Low-Light Optical Coherence Tomography Angiography Images under the Constraints of Vascular Network Topology"

_sensors, 2024, doi:10.3390/s24030774_

Round 1

Reviewer 1 Report

Comments and Suggestions for Authors

The authors have presented a Low-light OCTA images segmentation under the constraints of vascular network topology, which the proposed method is based on topological structure generation. The presented method is analyzed and verified by qualitative and quantitative metric analyses. Also, the provided paper introduces several innovative techniques aimed at advancing OCTA image segmentation and analysis.  The paper is well written and the idea is interesting; however there are some modifications in the manuscript, listed as follows, which should be corrected.

-        The impact of the proposed STA and BLAAM modules is discussed, but more detailed insights would be helpful; why these modules were chosen?

-        Kindly discuss how well the proposed model generalizes to different datasets or scenarios. If there are limitations in generalization, address them explicitly.

-        Kindly explain any specific characteristics or challenges in the RVOS and OCTA-500 datasets that make them suitable for this study. Explain why these datasets were chosen for evaluation?

-        Clearly state the motivation behind the proposed model. Why is the segmentation of OCTA images important, and what challenges does it address?

-        The conclusion is well written; However, the conclusion could be more concise and should explicitly restate the main objectives addressed in the study. Also, it should provide a more comprehensive summary of the key contributions, findings, and implications of the study.

Author Response

First and foremost, I would like to extend my heartfelt gratitude for the time and effort you have dedicated to reviewing our manuscript. Your insightful comments and suggestions are invaluable to us and have significantly contributed to the improvement of our work.

We have carefully considered each point you raised and have made corresponding revisions to the manuscript. To provide a detailed response to your comments, we have prepared a comprehensive reply which can be found in the attached file.

We believe that these revisions have substantially enhanced the quality and clarity of our paper, and we hope that our responses adequately address the concerns you have raised.

Once again, thank you for your valuable feedback and for aiding in the enhancement of our research.

Reviewer 2 Report

Comments and Suggestions for Authors

This paper proposes a new deep learning approach to improve the accuracy for the segmentation of Optical Coherence Tomography Angiography (OCTA) images. The proposed approach uses an architecture designed based on transitions from superficial to deep retinal layers. Experimental results on two datasets show that the proposed approach can achieve segmentation accuracy higher than that of several state-of-the-art segmentation methods. The paper is generally well written and the proposed approach is original and may have applications in clinical practice. However, the following issues need to be addressed before the paper can be accepted for publication.

1.       In Section 3, a clear description of the problem the paper intends to solve needs to be presented. For example, how many different classes of pixels need to be recognized by segmentation?

2.       In Section 3.1.1, a better explanation on Fl should be provided in the text. For example, how is the encoder that generates Fl’s designed? How many layers does it contain? etc.

3.       In Equation (1), Section 3.1.1, what are the values of H and W in all experiments?

4.       In Equation (5), Section 3.2.1, the mapping P_theta needs to be clearly explained in the text.

5.       In Equation (6), Section 3.2.2, the parameterized operator H_phi needs to be clearly explained in the text.

6.       In Section 4, some experimental data on the computational efficiency of the proposed approach should be provided.

Minor point:

In lines 53-54, page 2, some information might be missing in the sentence that starts with "such as However, ...".

Author Response

Dear Reviewer,

First and foremost, I would like to express my sincere gratitude for the time and effort you have dedicated to reviewing our manuscript. Your insightful comments and suggestions are invaluable to the enhancement of our work.

Please find attached our detailed responses to your comments and queries in the enclosed document. We have meticulously addressed each point raised and have made the necessary revisions to our manuscript accordingly.

Thank you once again for your constructive feedback and guidance. We are hopeful that our revisions meet your expectations and further strengthen the quality of our research.

Reviewer 3 Report

Comments and Suggestions for Authors

The main contributions of this study: Design of the PrIS-TS as part of the Biological information signal transduction imaging framework (BiSTIM);  Sub-Path structure constrain module (STA) that can provide deep supervision signals to enhance the priori of the segmentation structure; Bio-Luminescence Adaptation for Artifact Mitigation (BLAAM) that is used to alleviate imaging artifacts such as projection shadows; experimental evaluation of novel framework on two OCTA retinal vessel segmentation datasets: RVOS and OCTA-500 that demonstrates the effectiveness of proposed technique.

Comments:

      1)   This reviewer proposes for authors explaining all procedures of novel system (figs. 2-4) in form of the pseudocode algorithm where all operations can be mentioned. This permits better understanding of authors´ novel framework by potential reader.

2    2)  This reviewer suggests for authors to present the experimental setup (hardware used) used in their experiments (Sect. 4. Experiments and Results).

3   3)  The authors should present more information about chosen parameters in the experimental set. They wrote:” Our model was implemented using PyTorch, and all networks were trained using the Adam optimizer, with a learning rate set to 1 × 103 and weight decay of 1 × 106“. The chosen parameters values in Adam optimizer, as well as some other parameters of designed framework: tetta (eq.5), and fi (eq.6) should be explained.

Author Response

Dear Reviewer,

First and foremost, we would like to extend our sincerest gratitude for your valuable time and effort in reviewing our manuscript. Your insights and suggestions are immensely appreciated and have been instrumental in enhancing the quality of our work.

We have thoroughly addressed your comments and concerns in our revised manuscript. For your convenience, we have included a detailed response to each of your points in an attachment to this email. We believe that these revisions have significantly improved our manuscript, and we hope that you will find our responses satisfactory.

Once again, thank you for your constructive feedback and for contributing to the improvement of our research. We eagerly await your further comments and decision.

Reviewer 4 Report

Comments and Suggestions for Authors

This paper proposed “Low-light OCTA Images Segmentation Under the Constraints of Vascular Network Topology”. The approach discussed in this manuscript is interesting. To enhance the quality of the research, I recommend following corrections.

a)     Abstracts need to improve.

b)     Add the discussion section and future problems related to this topic.

c)     Conclusions are too small and need to improve.

d)     In the introduction section, cite some relevant articles.

e)      In the manuscript, there are many grammatical errors and typos. Carefully revised all manuscripts and corrected them.

Comments on the Quality of English Language

English editing required .

Author Response

Dear Reviewer,

First and foremost, we would like to express our sincere gratitude for the time and effort you have dedicated to reviewing our manuscript. Your insights and suggestions are invaluable to the improvement of our work.

We have carefully considered your comments and suggestions, and have prepared a detailed response to each point raised. This response, along with any corresponding revisions to the manuscript, has been compiled in a document which we have attached to this email for your convenience.

We hope that our responses and revisions adequately address your concerns and contribute to the enhancement of our manuscript. We eagerly await your feedback and are more than willing to make further adjustments if necessary.

Thank you once again for your invaluable contribution to our work.

Round 2

Reviewer 2 Report

Comments and Suggestions for Authors

All issues have been carefully addressed in the revised version. I have no other concerns and recommend the acceptance of the paper.

Reviewer 3 Report

Comments and Suggestions for Authors

The authors have attended all comments of this reviewer,